A giant specimen of Rhamphorhynchus muensteri and comments on the ontogeny of rhamphorhynchines

Hone David W.E. dwe_hone@yahoo.com 1
McDavid Skye N. 2
1 School of Biological and Behavioural Sciences, Queen Mary University of London , London , United Kingdom
2 Rye , NY , United States of America
Knoll Fabien
Electronic publication date: 2025 Jan 2
Publication date: 2025
Volume: 13
Electronic Location ID: e18587
Received 2024 Jun 19; Accepted 2024 Nov 4
Copyright: ©2025 Hone and McDavid
Copyright year: 2025
Copyright holder: Hone and McDavid
License: This is an open access article distributed under the terms of the Creative Commons Attribution License, which permits unrestricted use, distribution, reproduction and adaptation in any medium and for any purpose provided that it is properly attributed. For attribution, the original author(s), title, publication source (PeerJ) and either DOI or URL of the article must be cited.
License URL: https://creativecommons.org/licenses/by/4.0/

Keywords: Rhamphorhynchinae, Ontogeny, Pterosauria, ‘rhamphorhynchoid’, Solnhofen

Funding: The authors received no funding for this work.

==============================
Rhamphorhynchus is one of the best-known pterosaurs, with well over 100 specimens being held in public collections. Most of these represent juvenile animals, and the adults known are typically around 1 m in wingspan. Here we describe a near complete skeleton, preserved partially in 3D, of an animal with a wingspan of around 1.8 m, that is considerably larger than other known specimens, and is among the largest known non-pterodactyloid pterosaurs. This animal shows differences in the anatomy not seen in smaller specimens, revealing details of late-stage ontogeny in this genus. The specimen exhibits a disproportionate reduction in the size of the orbit and increase in the size of the lower temporal fenestra, a reduction in the proportional mandibular symphysis, and unusually laterally flattened teeth, which may point to a changing diet as these animals grew. These features show a transition from smaller to larger specimens of Rhamphorhynchus and also appear in other large specimens of rhamphorhyhchines and point to a consistent pattern in their development.

Introduction

Rhamphorhynchus is a genus of non-pterodactyloid pterosaur well known from the Solnhofen area Lagerstätten of southern Germany (Wellnhofer, 1975), although some partial remains have been referred to this genus from other European localities (e.g., O’Sullivan & Martill, 2015). It is widely regarded as an animal that foraged extensively in aquatic environments around the Solnhofen lagoons and was primarily piscivorous, based on numerous specimens preserved with fish as stomach contents (Witton, 2018), though other aquatic (Hoffmann et al., 2020) and perhaps even terrestrial prey were occasionally taken (Hone et al., 2015).

Due to the large amount of well-preserved material, it is represented by more complete specimens than any other pterosaur and is by far the best-known non-pterodactyloid taxon and the best-known outside of the Cretaceous. At least 125 specimens are present in public collections with others also recorded in private hands (Wellnhofer, 1975; Hone, Habib & Lamanna, 2013; Hone et al., 2020), and many of these are largely complete and articulated, if typically compressed into two dimensions, though a handful of acid-prepared specimens are informative for three-dimensional skeletal anatomy (CM 11431, CM 11434, NHMD 1891.738; Hone, Habib & Lamanna, 2013; Bonde & Leal, 2015). As a result, this taxon has been used extensively in numerous studies of different aspects of pterosaurian biology including neuroanatomy (Witmer et al., 2003), biomechanics (Witton, 2008; Persons & Currie, 2012), diet (Henderson, 2018; Bestwick et al., 2020), and particularly growth (Bennett, 1995; Prondvai et al., 2012; Hone et al., 2020).

Specimens range in size from an approximate total wingspan (here taken as the twice the length of the humerus, ulna or radius, wing metacarpal and all four wing phalanges combined) of approximately 0.3 m to 1.7 m. Most specimens have been considered osteologically immature based on their small size, unfused elements and coarse bone textures (Bennett, 1995), but medium sized and in particular larger specimens likely represent osteologically mature adults (Prondvai et al., 2012) (sensu Hone, Farke & Wedel, 2016). Adult pterosaurs show fusion of major elements such as the cranium and wrists, and exhibit an external fundamental system in the cortex of long bones and endosteal lamellae in the medullary cavities of long bones (see Kellner, 2015; Griffin et al., 2021). One specimen of Rhamphorhynchus, NHMUK PV OR 37002 is exceptionally large (Fig. 1), having skeletal elements of approximately one-third larger than the next largest known Rhamphorhynchus (GPIT RE/7321, termed ‘exemplar 81’ by Wellnhofer, 1975), which itself is considerably larger than other specimens.

NHMUK PV OR 37002 has had very little attention in the scientific literature to date. It was listed in Lydekker’s (1888) catalogue of fossils held at the NHMUK, and has a brief description by Woodward (1902) where he named this as a new species R. longiceps . It was later mentioned by Wellnhofer (1975) as ‘exemplar 82’ and Bennett (1995) as “the largest known specimen”, though he regarded it as simply an unusually large specimen of the monotypic genus. Bonde & Leal (2015) later suggested that it was indeed a separate species of the genus Rhamphorhynchus. However, it has never been discussed or illustrated in detail (Bonde & Leal, 2015 illustrated only the skull in ventrolateral view) in the literature, yet this specimen is potentially important for several reasons. First, it is preserved largely in three dimensions, which is rare for Solnhofen vertebrate specimens and thus provides rarely recovered information. Secondly, it is the largest known specimen of Rhamphorhynchus, which is important for understanding the growth of this taxon, especially at upper sizes. Finally, it is also among the largest non-pterodactyloid pterosaurs known, and certainly the most complete specimen of an animal in excess of 1.5 m in wingspan. Here we describe this specimen and show that contrary to some suggestions, it is not a distinct species, but is a member of Rhamphorhynchus muensteri and that it reveals a number of traits that developed late in ontogeny in large rhamphorhynchine pterosaurs.

Specimen history and locality information

According to the museum label that accompanies specimen NHMUK PV OR 37002, this formed part of the ‘Häberlein Collection’ and came from ‘Eichstädt’ [sic]. This specimen came to the museum as part of the 1862 purchase of Solnhofen specimens from Dr Karl Häberlein that also included the famous London specimen of Archaeopteryx (Lydekker, 1888). This would therefore be one of the specimens from the Eichstätt locality (this is also given by Woodward, 1902) and the Schernfeld-Eichstätt Basin, which is dated as Malm Zeta 2 and is therefore Tithonian in age (Bennett, 1995). Numerous Rhamphorhynchus specimens have been recovered from this locality, including all specimens that were previously referred to the species Rhamphorhynchus ‘longicaudus’ (Bennett, 1995).

Figure 1 Specimen NHMUK PV OR 37002 of a giant specimen of Rhamphorhynchus muensteri.

Arrow indicates the area recently reprepared (see also Fig. 4); counterplates and separate plate containing caudal series attached to the main plate are outlined in red. cdv, caudal vertebrae on separate plate and counterplate attached to main plate; hu, humerus; lwpx1, left wing phalanx 1; lwpx2, left wing phalanx 2; lwpx3, left wing phalanx 3 on counterplate; lwpx4, left wing phalanx 4 on counterplate; olwpx2, outline of left wing phalanx 2 on counterplate; olwpx3, outline of left wing phalanx 3; olwpx4, outline of left wing phalanx 4; orwpx3, outline of right wing phalanx 3; orwpx4, outline of right wing phalanx 4; orpes, outline of right pes; rad/uln, radius and ulna; rpes, right pes on counterplate; rwpx2, right wing phalanx 2; rwpx4, right wing phalanx 4 on counterplate. Scale is 5 cm with 1 cm increments.

Prior to this new description, the specimen was partly reprepared by Mark Graham, a senior preparator at the Natural History Museum, London. Although generally well preserved, the specimen was incompletely prepared and various parts had been repaired or supported with plaster or other materials. In 2016–2017, the specimen was mechanically reprepared, which revealed additional details of parts of the skull (especially the posterior face of the cranium), the cervical series, and the shoulder and chest region. The material was photographed in detail before the work began to document the specimen before the additional preparation was carried out (see photogrammetry model built using Agisoft Metashape software in the Supplemental File).

Description

Numerous specimens of Rhamphorhynchus have been described and illustrated in detail at various times and thus its anatomy is well known including both the skeletal system and soft tissues (e.g., Marsh, 1882; Wellnhofer, 1975; Bennett, 1995; Bonde & Christiansen, 2003; Frey et al., 2003; Hone, Habib & Lamanna, 2013; Bennett, 2015; Bonde & Leal, 2015). Although many specimens are compressed or crushed, a few specimens show remarkable three-dimensional preservation (e.g., see Frey et al., 2003; Hone, Habib & Lamanna, 2013; Bonde & Leal, 2015), and the number of specimens available means that most elements of the skeleton are known well as 3D structures. As a result, this description will focus on key traits of this individual (see Table 1 for various measurements of elements) rather than the already-described aspects of the species more broadly.

Table 1 Measurements of major elements of skeletal units of NHMUK PV OR 37002. All are taken to the nearest mm and were taken with callipers.

Element or unit	Length (mm)	
Skull (total length)	201	
Skull height (at quadrate)	49	
Skull width (across quadrates)	37	
Longest tooth (length)	20	
Mandible length (including reconstruction)	156	
Cervical (best preserved, maximum length, omitting plaster)	24	
Caudal series (minimum length)	462	
Scapula (length, to base of glenoid)	74	
Glenoid (anteroposterior length)	9	
Humerus (minimum and maximum length, omitting plaster)	70, 78	
Humerus diaphsysis (diameter anteroposterior and dorsoventral)	12, 6	
Humeral head (width)	23	
Radius (length, as exposed)	80	
Ulna (length)	103	
Wing phalanx 1 (length as preserved, left then right)	133 / NA	
Wing phalanx 2 (length, left then right)	168 / 176	
Wing phalanx 3 (length, left then right)	139 / 136	
Wing phalanx 4 (length, left then right)	136 / 137	
Femur length (as exposed)	44	
Demur diameter	7	
Metatarsals I-V	40, 41, 39, 33, 23	

NHMUK PV OR 37002 comprises most of a pterosaur skeleton including the skull and mandible; cervical, dorsal and caudal vertebrae; several dorsal ribs; both scapulocoracoids; virtually all major elements of the left wing; a partial right wing; a complete left hindlimb and elements of the right hindlimb. Degrees of articulation vary with some parts being in, or near, natural articulation, but others being separated from their original positions. The caudal series, right pes, and both wing fingers are articulated. The left humerus, radius, and ulna are articulated with each other but separated from the scapulocoracoid. The preserved cervical vertebrae are close to their life position. Only the thoracic region is separated from the skeleton. The specimen is preserved primarily in dorsal view with the skull in right lateral view.

There are a series of major breaks across the slabs on which the specimen is preserved, and several parts have been apparently moved and restored to places approximating a natural position — a practice seen in a number of restored Solnhofen region pterosaur specimens (see Hone, 2010). Similarly, the wing phalanges and the tail are mostly split between the main plate and counterplates, the latter of which have been attached to the main plate next to their counterparts (see Fig. 1). The specimen retains lots of plaster between elements, indicating considerable reconstruction before mounting into its wooden frame. Bones that cross breaks in the slabs — which include the cranium, mandible and both wing finger elements — are slightly distorted. However, the long axes of the bones align almost perfectly despite the complex nature of the break to the underlying matrix. For example, there is a total of eight mm in difference in length between wing phalanx II of the left and the right side (left is 167 mm), suggesting an imperfect, but reasonable repositioning in the slab.

Much of the specimen is preserved in three dimensions, although there is some crushing and damage to various parts. The skull is partly sheared such that the right side has been raised and the midline elements either are more raised than normal (e.g., the nasals), or more depressed (e.g., the postorbitals), which gives the skull a slightly unusual appearance (Fig. 2). The cranium and mandible show a division based on a major break along the anterior border of the orbit, and the jugal and lacrimal have been partially restored with plaster. As with the long bones mentioned above, although the join is imperfect, the general orientations, shape and lengths of the elements suggest that the skull has been reassembled accurately and the odd shape and appearance of the skull are due to its original taphonomic deformation and not due to the repositioning of different parts of the skull on the slab during repair.

Figure 2 Skull of Rhamphorhynchus muensteri NHMUK PV OR 37002 in near lateral view showing the 3D nature of the specimen (A) and restoration of the cranium and mandible in right lateral view (B).

Preserved bone and teeth are in white, obscured or reconstructed portions are in grey. Note the skull has no visible sutures. stf, supratemporal fenestra; ltf, lower temporal fenestra; orb, orbit; aof, antorbital fenestra; en, external naris; lwpx1, left wing phalanx 1; lwpx2, left wing phalanx 2. Scale bar is 50 mm.

Various elements are also split or are missing parts of the cortex, exposing the internal bone cavities. There is only limited evidence of calcite crystals on the specimen which are generally common on Solnhofen pterosaurs (Wellnhofer, 1975). The texture of the bone of the animal is smooth, indicating that it is not a juvenile, and major sutures (e.g., the wing extensor tendon process, between the scapula and coracoid, within the skull) are obliterated, indicting full osteological maturity (Bennett, 1995; Kellner, 2015; Griffin et al., 2021).

Skull

The cranium and mandible are near complete and articulated (Fig. 2). The right side of the skull, the dorsal part of the cranium and the ventral part of the mandible are all exposed. The posterior cranium is very partially exposed (the quadrates and occipital condyle are visible, but very poorly preserved), but other areas (in particular the palate) remain covered and could not be further exposed through preparation without risking damage to these fragile areas. Notably, the ventral margin of mandible sticks out and is intact, indicating resistance to crushing, however, the posterior part of the visible left mandible has been forced up into the temporal region when skull was crushed dorsoventrally.

There are ten alveoli in the upper jaw, with five teeth being preserved in them. The ten alveoli presumably represent six in the maxilla and four in the premaxilla, as is usual for the genus, although the suture between these elements cannot be seen. The anteriormost alveolus on the left is covered in matrix, but its presence is inferred based on a bulge in the jaw and the presence of a corresponding tooth on the opposite side. There is an apparent 11th tooth, but this is the anteriormost tooth from the left side of the jaw that protrudes between the right anterior teeth. Seven dentary teeth are inferred from swollen alveoli although only two of the more anterior teeth are present. The teeth are somewhat blunt at the tips and are also laterally compressed and thin, to the extent that the repreparation was halted to prevent damage to them.

The skull exhibits minor dorsoventral compression, with the nasals and frontals slightly displaced ventrally, making the skull roof appear concave rather than convex.

Axial skeleton

The axial skeleton appears to be generally in articulation based on the positions of visible elements and other parts of the skeleton, though only some cervical vertebrae and the tail are clearly visible. Much of the dorsal series and sacrum are not seen and may have been lost, or more likely, based on the otherwise complete nature and articulation of the specimen, are present but are not exposed.

At least two middle cervical vertebrae are partially exposed ventrally and one is also exposed in lateral view. The anteriormost of these three has some plaster infilling part of it, and all are partly covered by matrix and details are difficult to make out. There are two dorsal vertebrae preserved in transverse section (Fig. 3). As with a number of postcranial elements where the cortex is damaged, these show thin bone walls (approximately 0.3 mm) that are typical of pterosaurs. There are also two more dorsal vertebrae that are possibly fused to one another, but these are difficult to see as they are overlaid by dorsal ribs.

Figure 3 A dorsal vertebra of NHMUK PV OR 37002.

This is poorly preserved but is a previously unseen element, having been revealed by the new preparation work. cen, centrum; nc, neural canal; ns, neural spine; tvp, transverse process. Scale bar is 10 mm.

The caudal series is well preserved, though split between the plate and the mounted counterplate, down to the distal tail. The long chevrons and zygapophyses of the tail hamper our attempt to count the vertebrae, but there are at least 30 present. This does not include the tiny tip of simple caudals that are occasionally preserved in Rhamphorhynchus (e.g., see Hone et al., 2015) and these are not present here.

A small number of dorsal ribs are preserved in alignment, perhaps indicating some degree of articulation of the chest before burial. There is no evidence of the sternum or gastralia, although a number of smaller bone fragments associated with the torso are visible that may represent some of these elements.

Appendicular skeleton

Both pectoral girdles are preserved. The left scapulocoracoid is exposed in lateral view and overlies right, which is exposed in medial aspect. The coracoids of each are only partly exposed, with the left one being buried under the humerus. The left glenoid is poorly preserved, but mostly exposed and shows the typically ‘asymmetric’ configuration of rhamphorhynchines (Witton, 2015) with a poster oventrally positioned buttress, that prevents the humerus being positioned below the horizontal. The supraglenoidal buttress is confluent with the ventral margin of the scapula.

The left forelimb is the more intact of the two and comprises the humerus, radius and ulna, and a complete left wing finger. The carpals, wing metacarpal, metacarpals and phalanges are missing or more likely given the articulation, lie under the cervical series and are not exposed. The humerus is exposed dorsally and posteriorly, and being uncrushed, it allows for an unusually good appreciation of the three-dimensional shape of the bone. Woodward (1902) states that the humerus is incomplete but could not have exceeded ‘0.075 m’, although Wellnhofer (1975) gives this as 79 mm, and here we measure this as a maximum of 77.6 mm. The articular surface of the humeral head is gently arced and measures approximately 18 mm across (a portion of the dorsal region is missing). The ulnar crest deflects posteriorly from the posterior margin of the humeral head, though its exact morphology is obscured by matrix. The deltopectoral crest projects prominently from the diaphysis, tapering from a broad base to a relatively rounded termination. Some aspects of the terminal deltopectoral crest are difficult to establish given the current state of specimen preservation, but termination does not look swollen or ‘hatchet shaped’, as is often reported in ‘rhamphorhynchoid’ pterosaurs, including other Rhamphorhynchus (e.g., Wellnhofer, 1975; Unwin, 2003; Padian, 2008). The posterior surface of the proximal diaphysis contains a 9 mm long sediment-filled sulcus. We were unable to ascertain if this penetrates the bone cortex, but note that it is similarly positioned to pneumatic openings in some other pterosaur humeri (see Unwin, 2003). The diaphysis is gently bowed anteroposteriorly, and bears a muscle scar on its posterior surface. The supracondylar process is preserved adjacent to the broken and plastered distal end of the humerus. The breaks are sharp and imply that the humerus was complete as preserved, with the distal condyles lost during collection.

Only the proximal part of the right humerus visible, though it does allow the bone wall thickness to be measured on the dorsal surface of the humerus, and on the ventral surface of the deltopectoral crest, and are both approximately 0.6 mm thick (Fig. 4). Also preserved is a proximal radius, a possible ulna, a partially exposed wing metacarpal and metacarpals I–III. All wing finger elements are present for the right wing, but they are incompletely preserved and the proximal part of wing phalanx 1 is missing. There is a very slight curvature to the distal part of both wing phalanges 4, and both show a slightly expanded, rounded and ball-like distal tip which is seen in a number of pterosaurs, including other specimens of Rhamphorhynchus (Hone, Van Rooijen & Habib, 2015). Breaks to the bones means that the bone wall thickness can be measured here with some confidence — in wing phalanx 3, this can be measured at between 0.59 and 1.09 mm (Fig. 5), with the diameter of the element at this point being c. 8 mm.

Figure 4 The chest region of NHMUK PV OR 37002.

Photograph (A) and interpretive drawing (B) of the chest region of NHMUK PV OR 37002 after additional preparation. lwpx1, left wing phalanx 1; uln, ulna; rad, radius; ?, unknown; Lsc, left scapulocoracoid; rsc, right scapulocoracoid; hu, humerus; dpc, deltopectoral crest; r, rib; pat, pathology; dv, dorsal vertebra. The recently-prepared area is in the centre. The darker shaded area indicates where the specimen was reprepared. Scale bar is 100 mm.

Figure 5 Broken wing phalanx of NHMUK PV OR 37002.

Close up of the midshaft of a broken third wing phalanx on NHMUK PV OR 37002 showing the bone wall thickness. Scale in centimeters.

Most elements of the right hindlimb are present, and of the left hindlimb, only a few possible elements of the left foot can be identified that are exposed. The proximal end of the right femur is exposed, and this shows a large and well ossified femoral head which is somewhat flattened. The tibia is broken, and the middle part is either lost or not exposed. The distal end of the element is present, however there is extensive calcite crystal build-up over the tarsal region, and so little detail can be made out. The tibiotarsus is in articulation with the nearly complete right pes which lacks only the unguals. The foot is preserved well and the counterplate with impressions of these elements is also present with the specimen.

Discussion

Taxonomic identity

Woodward (1902) named the specimen as a new species, Rhamphorhynchus longiceps, and this identification and attribution is given with the specimen’s accompanying label. Woodward’s new species was based on extremely limited evidence, but is an available name under ICZN Article 12 (ICZN, 1999). He noted its large size and that its skull was proportionally long, compared to R. ‘gemmengi’ (NHMUK PV R 2786), though in fact this specimen has a skull in proportion with the rest of its body compared to other Rhamphorhynchs specimens (Hone et al., 2020). Woodward also noted that the toes were about half the diameter of those of R. ‘grandis’, based on a specimen at the NHMUK, though this specimen (NHMUK PV OR 42737) is clearly the pes of a large pterodactyloid because of the reduced 5th toe, and so such comparisons would not reveal anything about Rhamphorhynchus. No further comparisons were made to other than named species of Rhamphorhychus or defining traits listed.

Rhamphorhynchus has a complex taxonomic history with numerous species named at various times (e.g., see Wellnhofer, 1975). However, in a major revision of the genus, Bennett (1995) demonstrated that the previously suggested species actually formed several discrete year classes of both juvenile and adult animals that ultimately are from a single species — Rhamphorhynchus muensteri — an assignment that has been broadly adopted (e.g., Bonde & Christiansen, 2003; Unwin, 2003; Prondvai et al., 2012; Witton, 2013; Hone, Habib & Lamanna, 2013; Hone et al., 2020) and that we follow here. R. longiceps is therefore a junior subjective synonym of R. muensteri.

Bennett (1995) gave a thorough new diagnosis of this species, although Hone et al. (2012) showed that a number of these traits also overlap with the then newly identified rhamphorhynchine genus Bellubrunnus. NHMUK PV OR 37002 can be identified as R. muensteri based on the presence of the following traits (Bennett, 1995): 34 teeth (four in each premaxilla, six in each maxilla and seven in each dentary); anterior teeth long and angled forward and laterally; the fourth premaxillary tooth larger and more lateral than other premaxillary teeth, and posterior teeth shorter and more vertical. Two additional traits listed by Bennett (1995) — lower temporal fenestra narrow, upper temporal fenestra larger (than the lower) and rounded — may not be present here, as the lower temporal fenestra is not that narrow, and may be a similar size to that of the upper. However, the shape of the lower temporal fenestra is subject to individual variation in R. muensteri, even among similarly-sized specimens (compare e.g., CM 11431 and CM 11434 — see Hone, Habib & Lamanna, 2013).

Additional characters used by Bennett (1995) also appear in Bellubrunnus (Hone et al., 2012), but in this context, they are useful for diagnosing Rhamphorhynchus, as the former genus is from the Brunn locality, which is somewhat older than the other Solnhofen-type limestones (Hone et al., 2012), and NHMUK PV OR 37002 lacks autapomorphies that diagnose Bellubrunnus (e.g., only 22 teeth, absence of elongate zygapophyses on the tail). Therefore, additional traits of Bennett (1995) can also be used here to further support the identification of NHMUK PV OR 37002 as R. muensteri: jaws with edentulous tips, orbit substantially bigger than the naris and antorbital fenestra, the first wing phalanx is the longest and roughly the length of the skull. The two final characters given by Bennett (1995), femur shorter than humerus and prepubis slender and arched with a lateral process, cannot be confirmed because key elements cannot be observed.

Notably, Bonde & Leal (2015) retained R. longiceps as being a distinct species from R. muensteri based on a number of features and that they specifically state to be present in NHMUK PV OR 37002. These are: the temporal fenestrae being different in shape, “the upper more rounded and the lower wider than in the other forms”; different “size and proportions of the orbit in relation to the temporal openings”, the “upper jaw is not as pointed …and the lower jaw symphysis appears shorter, and the lower jaw is equally long as the upper”, and finally that “as reconstructed by Wellnhofer (1975), the fourth tooth is in the maxilla, not in premaxilla”.

However, our examination of the specimen suggests that this is not a strong set of traits for a referral of the specimen to a distinct species (Fig. 6). The lower fenestra here is wider than usually seen, but the upper does not appear to be any different in shape than seen in other specimens of Rhamphorhynchus (e.g., Wellnhofer, 1975 figure 3). The width of the lower fenestra may be a consequence of large size, and therefore represent an ontogenetic, rather than taxonomic difference (see below). Bonde & Leal (2015) do not state how the orbit in this specimen apparently differs to other specimens, which makes this suggested trait hard to assess. As they advocated that the other cranial openings are larger than normal, then the orbit would appear smaller as a consequence — this is effectively one trait and not two. Furthermore, since orbits in Rhamphorhynchus show negative allometry (Hone et al., 2020), then these will be proportionally largest in small animals, and proportionally shrink as they grow (even if absolute size still increases) with the largest individuals showing the smallest proportional orbits (Hone et al., 2020), a pattern common in vertebrates (Emerson & Bramble, 1993). As NHMUK PV OR 37002 is the largest known specimen, then proportionally small orbits relative to the cranium (and other fenestrae) are to be expected. Similarly, the upper jaw here appears to be just as pointed as other specimens of Rhamphorhynchus. The symphysis is difficult to assess as it is not clearly visible in many specimens, and these are mostly of similar sizes with no small specimens represented, however an assessment of a small number of specimens (see Table 2) does suggest that this proportionally shortens in this taxon as size increases (i.e., shows negative allometry). The upper jaw does not overhang as much in other specimens of Rhamphorhynchus, but also appears to be incomplete and so this is not a reliable difference. Finally, as noted above, the suture between the premaxilla and maxilla has been obliterated so there is no reason to think that the tooth counts in the two elements have changed, irrespective of how it may have been reconstructed by other authors. Numerous traits are shared with all other specimens of R. muensteri, and those that do differ are better explained as ontogenetic rather than interspecific differences. We therefore retain the referral of this specimen to R. muensteri. It should not be considered a separate taxon.

Figure 6 Different sized skulls of Rhamphorhynchus muensteri.

Skulls of specimens of Rhamphorhynchus muensteri at different sizes. Top to bottom: (A) BSPG 1889 XI 1 (‘Exemplar 7’, skull length 35 mm per Wellnhofer, 1975), scale bar 25 mm; (B) YPM VP 1778 (‘Exemplar 33’ of Wellnhofer, skull length 90 mm, measured by SNM using ImageJ), scale bar 35 mm; (C) GPIT RE/7321 (‘Exemplar 81’, skull length 150 mm per Wellnhofer, 1975, illustration mirrored and partially adapted from Wellnhofer, 1975), scale bar 50 mm; (D) NHMUK PV OR 37002, skull length 201 mm, scale bar 50 mm.

Size

NHMUK PV OR 37002 is considerably larger than other known specimens of Rhamphorhynchus (Fig. 7). The skull and humerus are respectively 201 mm and 78 mm in length, with the next largest specimen of R. muensteri (GPIT RE/7321, Wellnhofer, 1975 specimen number 81) having a skull of 150 mm and a humerus of 65 mm. Even this individual is much larger than most others, and there are a cluster of 12 specimens with skull lengths around 120-125 mm and humeri of 40-43 mm in length (see data in Habib & Hone, 2024). So NHMUK PV OR 37002 is more than 60% larger than all but one of the largest known Rhamphorhynchus specimens, and is the largest by around 33%. In contrast, the smallest specimen we know of (BMMS 3A) had a skull of 21 mm and a humerus of just 15 mm.

Woodward (1902) considered the size of the specimen notable, stating that NHMUK PV OR 37002 was ‘a distinct species…, larger than any hitherto discovered’, and it would also have been the largest non-pterodactyloid pterosaur known from the Jurassic at that time. There are large specimens of Dorygnathus (SMNS 81205 has a skull of 150 mm and humerus of 78 mm) and the skulls of Angustinaripterus (192 mm skull — He, Yang & Su, 1983) and Parapsicepahlus (140 mm — O’Sullivan & Martill, 2017) are large, but this Rhamphorhynchus remains of exceptional size (Fig. 7). One large specimen of Dimorphodon (NHMUK R 4121) has a comparable skull length (c. 220 mm) and even longer humerus (c. 90 mm), though the wingspan overall is considerably smaller than that of NHMUK PV OR 37002. The recently-described Dearc from Scotland has a skull of c. 220 mm and humerus of 112 mm, with an estimated wingspan of 2.5 m (Jagielska et al., 2022), while isolated axial and appendicular elements of an indeterminate non-pterodactyloid from the same Lealt Shale formation indicate an animal of even larger size (Jagielska et al., 2023). Some isolated wing elements from Solnhofen pterodactyloids also point to animals of approximately 2 m in wingspan (Elgin & Hone, 2020), and there are very partial specimens that suggest animals of 5 m in wingspan from the UK (Etienne et al., 2024), but overall NHMUK PV OR 37002 would have been one of the largest pterosaurs prior to the Cretaceous. Based on Witton’s (2008) relationship between mass and wingspan of non-pterodactyloid pterosaurs, the mass of NHMUK PV OR 37002 can be estimated as 3.5 kg.

Table 2 Measurements of the mandible (total length in the midline) and symphysis of specimens of Rhamphorhynchus.

Measurements are made to the nearest mm and were taken with callipers except the SMNK specimen which was taken with ImageJ from a photograph.

Specimen	Mandible (mm)	Symphysis (mm)	Percentage	
NHMUK PV OR 37002	156	43	28	
NHMUK PV OR 37003	89	36	40	
NHMUK 43004	78	29	37	
NHMUK R 2786	70	31	43	
NHMUK R 231	70	30	43	
SMNK PAL 6596	61	27	44	

Figure 7 Size comparison of different specimens of Rhamphorhynchus.

Size comparison of different Rhamphorhynchus muensteri specimens: (anti-clockwise from top left) the smallest known BMMS A3 (21 mm skull length), a generalised ‘typical adult’ specimen (122 mm skull length), the second largest known GPIT RE/7321 (150 mm skull length) and the largest known NHMUK PV OR 37002 (201 mm skull length). Scale bar is 1 metre.

Comparisons to smaller specimens of Rhamphorhynchus and other large rhamphorhynchines

Despite the large size of the specimen, proportions of various parts of the skeleton (including the head and wings) still fit with the near-isometric general patterns seen throughout the species from the smallest to largest specimens (Bennett, 1995; Hone et al., 2020), and the overall intraspecific variation in the species is low (Habib & Hone, 2024). This is again consistent with the specimen being part of R. muensteri. NHMUK PV OR 37002 shows a number of anatomical features that mark it as apparently unusual, compared to most specimens of Rhamphorhynchus. Despite the original species designation of ‘longiceps’, Woodward (1902) correctly noted that the edentulous rostrum of the specimen is short and deep, compared to most other specimens. In contrast, the mandible as a whole is mediolaterally narrow as the length to width ratio is around 4:1, which contrasts with a ratio of c. 3:1 on another Rhamphorhynchus specimen (NHMUK PV R 2786) of about half the absolute size. We suggest that as the symphysis fuses during ontogeny, this length could be reduced in adults, as while proportionally smaller in length, it would be absolutely bigger and stronger as it fuses and obliterates the suture. Notably, this change does not occur in at least one other early pterosaur. Dorygnathus shows a consistent symphysis length of c. 33% of the lower jaw length (measured with ImageJ from Padian, 2008, his figures 8 & 10 and plates 1, 4 and 5).

The lower temporal fenestra is considerably expanded and trapezoidal compared to smaller specimens of Rhamphorhynchus, both in height (28 mm) and midheight anteroposterior length (17 mm, maximum width of 18 mm), and is not the slit-like opening more usually seen in this genus (Fig. 6). This is apparently due to the postorbital bar being rotated forwards into a more vertical position, such that the dorsal end of the lower temporal fenestra is more open, and the orbit has a straighter posterior margin. This size and shape change may be a trajectory for larger non-pterodactyloid pterosaurs generally. We note that there is a similar, if less exaggerated, change in the lower temporal fenestra seen between smaller and larger specimens of Dorygnathus (Padian, 2008), and the large rhamphorhynchines Parapsicephalus (O’Sullivan & Martill, 2017) and Angustinaripterus show a similarly shaped fenestra (He, Yang & Su, 1983) (Fig. 8). The width of the skull at the exoccipitals is proportionally wider in NHMUK PV OR 37002 than seen in smaller specimens of the species. The posterior part of the skull is visible in posterolateral view in NHMUK PV OR 37002 (unlike in the vast majority of Rhamphorhynchus specimens), and here it is possible to see that there is no expansion/enhancement of exoccipitals despite the expanded width of the skull (c.f. Wellnhofer, 1975 his figure 2), although they are not well exposed or preserved and details are difficult to make out. However, the right quadrate is robust, and the medial expansion with the squamosal is dorsoventrally broad (Fig. 9).

Figure 8 Variations in the structure of the posterior skull in derived non-monofenestratan pterosaurs.

Posterior part of skulls of large non-monofenestratan pterosaurs showing their temporal fenestrae: (A) Dorygnathus banthensis SMNS 55886 after Padian (2008) and Wellnhofer (1978); (B) Angustinaripterus longicephalus ZDM T8001 after He, Yang & Su (1983); (C) Parapsicephalus purdoni GSM 3166, mirrored after O’Sullivan & Martill (2017); (D) Rhamphorhynchus muensteri NHMUK PV OR 37002; (E) Rhamphorhynchus muensteri YPM VPPU 11984 (exemplar 70 of Wellnhofer); (F) Dearc sgiathanach NMS G.2021.6.1-4. Dotted lines represent reconstructed parts of the skull. Specimens not to scale.

Figure 9 Posterior cranium of NHMUK PV OR 37002.

Posterior view of the cranium of NHMUK PV OR 37002, Rhamphorhynchus muensteri. exoc, exoccipital; lqt, left quadrate; occ, occipital condyle; rqt, right quadrate; rsq, right squamosal; socc, supraoccipital. Scale bar is 50 mm.

The teeth in NHMUK PV OR 37002 are particularly unusual as they are clearly somewhat laterally compressed (as also noted by Woodward, 1902) and contrast with the apparently subcircular cross-section of teeth that is typical in Rhamphorhynchus. The largest preserved tooth is over 19.5 mm in length, 6 mm across the base, but only approximately four mm thick, with the anteriormost premaxillary tooth being approximately 15 mm by 5 mm by 2.5 mm, respectively. Wellnhofer, (1975, his Fig 4) illustrates the teeth as being sub-oval in cross-section and examination of a number of specimens shows that they do not typically have subcircular teeth, but that these are at least a little laterally compressed. Although the preserved teeth and alveoli in NHMUK PV OR 37002 are unusually elliptical and flattened (Fig. 10), this may again be an exaggeration of a condition that was already present in Rhamphorhynchus and not observed before, as the diameter of teeth are very hard to measure. For example, the adult-sized (skull length of 95 mm) NHMUK R 2786 certainly appears to have more flattened teeth than smaller specimens, and this is also a feature seen in the teeth of Dearc (DWEH pers. obs.) and the anterior teeth of Angustinaripterus are described as being elliptical in cross-section with the posterior ones being laterally compressed (He, Yang & Su, 1983).

Figure 10 Flattened teeth of NHMUK PV OR 37002.

The anterior skull of NHMUK PV OR 37002 in ventrolateral view showing the relatively flattened teeth that are oval and not circular in cross-section. Scale bar is 50 mm.

Implications

That this specimen fits with the overall isometric pattern of much smaller individuals of the species is perhaps unusual, given that biomechanical factors, such as wing area, will increase at the second power, while mass will increase at the third power. Thus, various features such as the lengths of the humerus, or the wing as a whole might be expected to change at larger sizes to accommodate the shifts in various proportional forces, but this does not appear to be the case (see also Habib & Hone, 2024). We note for example also that the posterior expansion of the joints in the wing finger elements are similar to those of even much smaller specimens (e.g., NHMUK OR 2786), suggesting similar safety factors and associated forces on them. The deltopectoral crest lacks the restriction at the base, which is also seen in other large specimens of Rhamphorhynchus, though is present, or even exaggerated, in smaller ones. The additional or changing relative forces associated with an animal in proportion but at greater size may be offset by factors such as increased pneumaticity in larger animals or a fundamental change in flight pattern, but it is still notable how consistent the general patterns are for larger specimens, compared to even the smallest ones that have one fifth or less of the wingspan.

These changes in ontogeny in both cranial and tooth shape and the potential for different flight profiles means that NHMUK PV OR 37002 probably differed in its ecology compared to smaller specimens of Rhamphorhynchus. The change to the lower temporal fenestra and expansion to the back of the skull, coupled with a proportionally thin jaw, labiolingually narrower teeth, reduced mandibular symphysis relative to the length of the jaw, and shorter rostrum all point to a difference in feeding, be it prey type or method of acquisition and processing. Dietary shifts must have happened during ontogeny (Hone et al., 2020) and there is some evidence for this in pterosaurs, including Rhamphorhynchus, where it is suggested that they shift from a more insectivorous to more piscivorous diet during growth (Bestwick et al., 2020). However, given the diversity of diet known and inferred for Rhamphorhynchus (see e.g., Hone et al., 2015; Witton, 2018; Hoffmann et al., 2020) and at the largest sizes they may have shifted to still other prey, or had a different preferred prey types.

The increased posterior part of the skull with a short rostrum would suggest an animal with an absolutely more powerful bite (e.g., see Walmsley et al., 2013), but this is an odd combination with proportionally (though not absolutely) thinner teeth and a weaker jaw. A shift to more laterally compressed teeth would increase their ability to cut at the expense of being able to grab and swallow (D’Amore et al., 2024; Bugos & McDavid, 2024; but also see Rieppel, 1979; D’Amore & Blumenschine, 2009), and so may suggest that these largest rhamphorhynchines were less reliant on fish and similar prey (e.g., soft-bodied cephalopods) as part of their diet, or were using this cutting ability to process larger items (e.g., terrestrial tetrapods) into pieces that could be swallowed (compare to Bugos & McDavid, 2024). If so, this may also partly explain the rarity of larger animals if they tended to forage in more terrestrial environments and therefore were less likely to die and be buried in the local lagoons compared to aquatic foraging juveniles and small adults (see also, Bennett, 2018 on preservational potential and foraging in juvenile pterosaurs). Notably, however, the angle of the quadrate does not seem to change in ontogeny, being around 130-140 degrees in both small and large specimens (see Bennett, 1995, his figure 5) and is a similar value here. This suggests that the adductor muscles are not changing dramatically in their delivery of force during this growth, despite the other changes noted.

It has also been suggested that the hatchet shape of the deltopectoral crest seen in many pterosaurs (and including small specimens of Rhamphorhynchus, but not here) is linked to the ability to launch from water (Cunningham & Habib, 2011). Thus, these changes here may point to large animals being less reliant on feeding in aquatic systems, on fish and similar foodstuffs, and are instead now foraging for alternate prey in different environments. This would also then point to ontogenetic niche partitioning with adults and juveniles targeting different prey items.

This overall pattern may be true of other large animals that have been described as rhamphorhynchines. The large Dearc is from an estuarine locality (Jagielska et al., 2022) while Angustinaripterus is from the Xiaximiao (Shaximiao) Formation (He, Yang & Su, 1983), which is a fundamentally terrestrial system encompassing a floodplain (Xie et al., 2023). As such, rhamphorhynchines may have moved inland as they reached larger sizes and, while still tied to water bodies, have been more generalist feeders. As large adults, they would perhaps be analogous to some modern gulls (Laridae) — generalists who typically prefer marine or at least aquatic systems, but are capable of foraging successfully in more terrestrial systems.

Rhamphorhynchus (Bennett, 1995) and some other Solnhofen pterosaurs (e.g., Pterodactylus, Bennett, 1996) are unusual compared to most other tetrapods in that there are numerous juveniles represented and relatively few adults. As a result, although NHMUK PV OR 37002 was clearly much larger than other known specimens with relatively few of the c. 130 specimens known being of adult size, the sample here is effectively much smaller. Therefore, while NHMUK PV OR 37002 is a giant individual in terms of its absolute size compared to the rest of the species, it was perhaps not that much larger than others that are missing as a consequence of limited sampling, and especially if larger animals were foraging in more terrestrial environments.

NHMUK PV OR 37002 (Fig. 11) is an important specimen for understanding the growth and ecology of this species. Despite the large number of specimens known, individuals of outstanding size are an important indicator of what was mechanically and ecologically possible for non-pterodactyloids pterosaurs.

Figure 11 Reconstructed skeletal diagram of a giant specimen of Rhamphorhynchus.

Skeletal diagram of an osteologically mature Rhamphorhynchus muensteri based mostly on NHMUK PV OR 37002, missing parts modified from Witton (2013: 129) and Wellnhofer (1978) and scaled following Hone et al. (2020). Scale bar is 250 mm.

Supplemental Information

Supplemental Information 1 3D photogrammetric model of NHMUK PV OR 37002 before repreparation

A photogrammetric model of NHMUK PV OR 37002 in 3D taken before the recent repreparation work to document the state of the specimen before it was worked on.

Supplemental Information 2 List of specimens and museum repositories

Thanks to Sandra Chapman, Susannah Maidment and Mike Day for access to the specimen and Natalia Jagielska for access to Dearc. We thank Dan Brinkman and Vanessa Rhue for information and photos of YPM specimens. We thank Andrew Knapp for help with the photography and photogrammetry of the specimen. Special thanks to Mark Graham for his extraordinary work on the new preparation of the specimen, and to Mark Witton who made a major contribution to the early part of this project. Our thanks to three referees and the editor for their comments which helped improve the paper.

Institutional Abbreviations

BMMS Bürgermeister-Müller Museum, Solnhofen, Germany

BSPG Bayerische Staatssammlung für Paläontologie und Geologie, München, Germany

CM Carnegie Museum of Natural History, Pittsburgh, Pennsylviana, USA

GPIT Paläontologische Sammlung der Universität Tübingen, Tübingen, Germany

GSM British Geological Survey Museum, Keyworth, UK

NHMD Natural History Museum of Denmark, København, Denmark

NHMUK (formerly BMNH) Natural History Museum, London, UK

NMS National Museums Scotland, Edinburgh, UK

SMNK Staatliches Museum für Naturkunde Karlsruhe, Karlsruhe, Germany

SMNS Staatliches Museum für Naturkunde Stuttgart, Stuttgart, Germany

YPM Yale Peabody Museum, New Haven, Connecticut, USA (VPPU designates vertebrate paleontology specimens formerly held at Princeton University, Princeton, New Jersey, now held at YPM)

ZDM Zigong Dinosaur Museum, Zigong, Sichuan, China

Additional Information and Declarations

Competing Interests

Author Contributions

Data Availability

David W.E. Hone is an Academic Editor for PeerJ.

David W.E. Hone conceived and designed the experiments, performed the experiments, analyzed the data, prepared figures and/or tables, authored or reviewed drafts of the article, and approved the final draft.

Skye N. McDavid performed the experiments, analyzed the data, prepared figures and/or tables, authored or reviewed drafts of the article, and approved the final draft.

The following information was supplied regarding data availability:

The specimens are available at: BSPG 1889 XI 1, BMMS 3A , CM 11431, CM 11434, GPIT RE/7321, GSM 3166, NHMD 1891.738, NHMUK R 231, NHMUK R 2786, NHMUK PV OR 37002, NHMUK PV OR 37003, NHMUK 4121, NHMUK PV OR 42737, NHMUK 43004, NMS G.2021.6.1-4, SMNK PAL 6596, SMNS 55886, YPM VP 1778, and ZDM T8001.

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
