# Peer review of "A giant specimen of Rhamphorhynchus muensteri and comments on the ontogeny of rhamphorhynchines"

_PeerJ, doi:10.7717/peerj.18587_

## Round 0.1 · original submission · Major Revisions

I have now received three reviews of your manuscript submitted to PeerJ. I find many of the reviewers' comments to be valid, so I have decided that 'Major Revisions' are necessary. Please consider changing the title; I feel that the adjective 'giant' is not appropriate. How about 'large' or 'adult' instead? The paragraph that starts on L65 should be modified to include a reference to Bonde and Leal. There a number of typos to be corrected: L319, "is" after "jaw", L355 "asuggest" etc .

·

Basic reporting

Below are my main review points for the manuscript according to PeerJ’s recommended format. Some comments may be applicable to more than one subsection of the review. I have uploaded a fully reviewed PDF of the manuscript with all of my small- to medium-level comments and questions that relate to these three subsections using the sticky note and markup tools as I find this a more efficient way of reviewing.
1. Basic Reporting.
• As far as I am aware, Figures 3, 5 and 7 are not cited anywhere in the main text and so this must be rectified (easy to do).
• The authors do a good job of outlining why this specimen is important for understanding of the morphology and ontogeny of Rhamphorhynchus. However, a lot of the specimen’s taxonomic history (e.g. originally being categorized as R. longiceps) is not mentioned until the discussion when it would be better placed in the ‘specimen history and locality information’ subsection of the methods. As it stands the layout of the MS is a little disjointed.
• I appreciate the effort that went to creating the 3D model of the specimen before repreparation work was carried out in case anything bad was to happen to the specimen. However, the model that exists in the supplementary information now no longer reflects the current status of the specimen, given that new anatomical information is now available from it. Wouldn’t it make more sense to have a 3D model that shows the new anatomical information? Furthermore, there is no mention anywhere in the manuscript about how the 3D model was created and so as it currently stands is not reproducible (this also corresponds to the experimental design subsection of the review).
• The authors mention several times that the quadrates, exoccipitals and occipital condyle are partially exposed but are not featured in the figures. Given how much you state this I think they should be figured.

Experimental design

2. Experimental design
• One of the main aims of the manuscript, stated at the end of the introduction, is to “show that contrary to some suggestions, it [the specimen] is not a distinct species but is a member of Rhamphorhynchus muensteri…”. However, this is the first time in the introduction that the validity of the specimen as a R. muensteri has been called into question and so is not really set up that well. The taxonomy of the specimen is not mentioned or discussed properly until the discussion and even then the authors only cite one paper that hypothesizes the specimen’s taxonomic identity (Bonde & Neal 2015). As it stands, these parts of the manuscript unfortunately come across as weakly justified.
• The authors provide new measurements of the specimen based on the additional prep work that has taken place on the specimen. However, they do not state how these measurements were taken e.g. callipers vs ImageJ. This is an easy enough thing to add for transparency.
• Furthermore, some of the measurements given in this paper contrast with the original measurements made by Wellnhofer (1975). For example, the skull length differs by around 9 mm between the two studies. Are such differences due to the additional prep work that has taken so? If so, please say so because at the moment this is not clear.
• See above for my concerns about the 3D model provided in the supplementary information.
• A large portion of the discussion focuses on measurements of anatomical features of the skull between the study Rhamphorhynchus and smaller-sized specimens to understand ontogenetic changes in skull morphology. Such traits include orbit diameter relative to skull length, mandible height versus width and symphysis length relative to mandible length. However, the manuscript does not state that this has been done until the discussion where the authors state these measurements and compare between the differently sized specimens. There is also no mention about how such measurements were made (callipers in person or ImageJ from photos?). There should be designated sub sections in the methods and results for these data which would greatly improve the flow of the manuscript and to increase transparency.

Validity of the findings

3. Validity of the findings
• My main concern, perhaps of the entire manuscript, is the robustness of the arguments made about Rhamphorhynchus skull characteristics changing with ontogeny. As mentioned above, the authors looked at several anatomical traits such as orbit diameter relative to skull length, mandible height versus width and symphysis length relative to mandible length. However, the authors make unjustifiable conclusions about ontogenetic changes in Rhamphorhynchus skull anatomy based on extremely limited sample sizes. For example, the authors state that symphysis length relative to mandible length decrease with increasing size (negative allometry) but this is based on only four specimens, including the main study specimen. Mandible height versus width is only based on two specimens in total. The authors do not state the size or relative ontogenetic stages of these few comparative specimens, nor do they provide figures of them in what is largely a comparative based study. As it stands this is not scientifically robust. Larger sample sizes with quantitative statistical testing (allometric regressions) must be done in order to make logical interpretations on allometric scaling of skull characteristics.

Additional comments

No comment

Reviewer 2 ·

Basic reporting

The description is thorough and sufficiently illustrated, especially the add-on photogrammetry file. Albeit, to make the paper better adapted to act as a future anatomical reference, I have some minor recommendations to add:

"NHMUK PV OR 37002 has never been discussed or illustrated" There is a rudimentary sketch in Bonde & Leal (2015) (noted further in paper); "contrary to some suggestions" by who? "edentulous rostrum of the specimen is proportionally short and deep compared to most other specimens" might benefit from a reference to Klobiodon with a similar profile. "of larger animals if they tended to forage in more terrestrial environments" can be further corroborated with appearance of Harpactognathus and Sericipterus from terrestrial deposits. Noted above are suggestions and non-essential to the flow and delivery of the paper.

The figures provided are sufficient, but areas of interest, like high-resolution images and drawings of regions like the caudal vertebra or pes would benefit the paper, given it is the first detailed description of the specimen. I know it is hard to provide illustrations of 3D elements but given the importance of this specimen and the aim of this paper to be descriptive, this will improve the paper. Close up images too are of relatively low quality.

Experimental design

It would be of benefit to this paper to comparatively see illustrations or photographs of other large Rhamphorhynchus specimens, such as the Wellnhofer sp.81 (mentioned throughout in the text). There is a comparative size outline, but a comparative diagram of to-scale lateral skull and humerus will help with the visualisation of features and future reference.

The WP3 has a wing phalange bone wall thickness described, for values to be useful, radius of the bone and overall bone thickness would put the values into comparable context (i.e comparative ratio).

The inclination and morphology of the quadrate play a big part in the bite force, paralleling the adductors, but is sparling mentioned in the text, does it change with ontogeny in Rhamphorhynchus or remain a constant?

Another non-pterodactyloid of which we have a good sample size is Dorygnathus. Does the symphysis, dentition or other cranial changes also apply to Dorygnathines or are they only a Rhamphorynchus-ontogenetic sequence feature? In the paper lower temporal fenestra is of interest, but more features could be compared to note probable parallels (or lack thereof). This might help us in the future when looking at other non-pterodactyloids and deciding if Rhamphorynchus is a clade-wide model for inferring maturity.

Validity of the findings

This is a very good paper on a keystone Jurassic specimen is sure to be used as a reference for palaeontologists to come. The paper does not need statistical backing, as the legwork was already done in a preceding paper ("Unique near isometric ontogeny in the pterosaur Rhamphorhynchus suggests hatchlings could fly" by the first author).

I only have minor comments and suggestions of action,

Additional comments

Suggestions of further action, might be too time consuming and unreasonable, so remain as recommendations rather than required changes:

Suggestions. I know it is far removed from the feasibility of this paper. Still, this specimen would benefit so much from a CT scan, especially to unravel the palatal, endocast and full humeral shape which would help to narrow down its place in phylogeny and improve our understanding of the behaviour and anatomy of adult non-pterodactyloids. Like for all Sonhofen specimens, a UV photograph would help.

·

Basic reporting

This paper is interesting and should obviously be published with a few improvements and corrections.
A few words I cannot find in my Eng-Dan dictionary nor in Concise Oxf. Dictionary. Lin 152: ...."mandible is proud and is intact" - what does "proud" mean - is it usual language ?? Misprint line 319: .... upper jaw is not does not overhang as much ...

Line 106 they claim that the 3-D fossil "will not be described in detail, but focus on key traits" and refer to measurements. This is not reasonable, as so few specimens are preserved wih skulls in 3-D, and referal to a certain species and disagreement with others about this depends on rather small details.
So why not compare some details directly with our Danish acid prepared specimen of Rh. muensteri (by Bonde & Christiansen 2003, Bonde & Leal 2015 on vertebrae resp. skull - both being referred to [lin. 101-102], but not used the relevant way). This is probably the most informative specimen existing (only one more acid prep in Carnegie Museum - they must know that as Hone has published with Lamanna, C M) - none in German coll (I have seen many Rh. in Munich, Eichstätt, Stuttgart, Frankfurth, Berlin, local Solnhofen museums, London, Paris, Stockholm, New York, Carnegie Mus - and obviously I do not know how much they have seen). And our skull is not incl. in fig. 8 of posterior skulls.
With our detailed anatomical studies it is rather frustrating not being used in the proper way.

Experimental design

As it is a NHM specimen, why not CT-scan to look for several bones said to be either missing or buried in the rock ?? (There is even a paper during N-Amer Pal Conv, June 2024 by Atterly, Friedman, Z Johanson [NHM] & Giles about CT-scan presumably in NHM !!). E. g. line 200-202 left `hand` (hidden) under cervic. vertebrae - line 208-209 ulna. crest obscured by matrix.
So here is something lacking methodologically, espec as they claim (line 129-132) the skull being sheared and crushed to look "unusual" - and the palate cannot be seen (hidden). Line 187 no evidence of sternum.
Ad axial skeleton line 174-176: two mid cervic vertebrae seen ventral and lateral - and two dorsal vert. in transverse section (fig.3 NOT mentioned !) - they should be able to see some of the pneumatic foramina described in great detail by NB & Christiansen - and such is of great interest in ontogenetic studies.
Even in the poorly preserved dorsal vert. in fig 3 it should be possible to see some details - and CT-scan would probably help a lot. Fig. of the cervicals would be nice !

Validity of the findings

The idea of an ontogenetic series seems well founded and explained with 37002, Rh. longiceps being the largest Rh. muensteri. Although one should remember, that being one or two spp. here depends on the subjective evaluation of very small differences, like in their criticisme (line 288-303) of NB & Leal for accepting two spp. based on proportions of foramina in the skull. They argue that such diff. are age dependant - they may be right and show fig. 8 (without Danish specimen), but the arguments seem rather subjective as often in palaeontology when discussing differences at the species level. (I have myself studied this Rh. longiceps in London - I must admit finding the weight of 3.5 kg surprisingly low; line 360).
Under Taphonomy: Hone et al 2012 show overlap with Bellabrunnus, which is from loc. Brunn, just a little bit older than the other Solnhofen locs - and here they claim generic difference !
All the implications (line 426 ff) about feeding and environments as usual in palaeontol. are interesting and fancifull: from mechanics, shorter jaws, sharper teeth to more powerful bite that can cut easier, to shift in diet from fish and soft cephalopods (is the latter known from beaks in stomach content ?) to larger prey (terrestr. tetrapods) to cut it in small pieces, and not lagoonal, that seems very fanciful (but not impossible considering large birds today like storcks, marabou, owls, seriema etc) - but about gulls, contra their claim line 464, the largest are found at the coast.

Additional comments

A remark more about style (perhaps to be decided by editors): It is iritating and a bit disturbing with this very long museum-number - why not state in the beginning, that it will just be called "no. 37002" ?? (or some other short version).

---

## Round 0.2 · Minor Revisions

Please carefully address Reviewer #3's comments. Take the time needed to refine your text, including proper use of italics (lines 71, 268 etc) and other formatting adjustments.

·

Basic reporting

.

Experimental design

.

Validity of the findings

.

Additional comments

I am disappointed, that the authors will not use our skull illustrations from Bonde & Leal and those of pneumatizations from Bonde & Christiansen. Some of the latter they should really be able to see in their vertebra (and in birds they develop and grow fairly late - therefore important for ontogeny studies). As our acid-prepared specimen is about the specimen with most details in the world.

To get more details, the authors should have scanned the specimen, which could be easily done in NHM.

I think the discussion of the openings in the skull is not very satisfying, but they may still be right.

There is a lot of mistakes concerning italics, both concerning generic names, and differences between entire long passages:

"Implications" - the 18 line part is in italics (why), then that suddenly stops.
And not easily understandable sentences: under "symphyses": ..... size increases (i.e. shows negative allometry). meassured with callipers or unusual in ..... (And shortly after:) .... There is a general trend for -The upper jaw ..... ????

Litt: Bennett 1996 - in wrong place.

I still think the specimen no. should be shortened all through.

---

## Round 0.3 · accepted · Accept

I confirm that your paper is now accepted for publication.